# Joint Entropy Search for Multi-Objective Bayesian Optimization

**Ben Tu**[†]    **Axel Gandy**[†]    **Nikolas Kantas**[†]    **Behrang Shafei**[‡]
[†]Imperial College London
[‡]BASF SE
ben.tu16@imperial.ac.uk

## Abstract

Many real-world problems can be phrased as a multi-objective optimization problem, where the goal is to identify the best set of compromises between the competing objectives. Multi-objective Bayesian optimization (BO) is a sample efficient strategy that can be deployed to solve these vector-valued optimization problems where access is limited to a number of noisy objective function evaluations. In this paper, we propose a novel information-theoretic acquisition function for BO called Joint Entropy Search (JES), which considers the joint information gain for the optimal set of inputs and outputs. We present several analytical approximations to the JES acquisition function and also introduce an extension to the batch setting. We showcase the effectiveness of this new approach on a range of synthetic and real-world problems in terms of the hypervolume and its weighted variants.

## 1   Introduction

Bayesian optimization (BO) has demonstrated a lot of success in solving black-box optimization problems in various domains such as machine learning [76, 77, 91], chemistry [27, 35], robotics [7, 14] and clinical trials [63, 79]. The procedure works by maintaining a probabilistic model of the observed data in order to guide the optimization procedure into regions of interest. Specifically, at each iteration the black-box function is evaluated at one or more input locations that maximizes an acquisition function on the model. Implicitly, this function strikes a balance between exploring new areas and exploiting areas that have been shown to be promising. In this work, we consider the more general problem, where the black-box function of interest is vector-valued. This increases the difficulty of the problem because there are now many directions in which the objectives can be improved, in contrast to the single-objective setting where there is only one. Informally, the end goal of multi-objective optimization is to identify a collection of points that describe the best trade-offs between the different objectives.

There are several ways to define an acquisition function for multi-objective BO. A popular strategy is random scalarization [51, 64], which works by transforming the multi-objective problem into a distribution of single-objective problems. These approaches are appealing because they enable the use of standard single-objective acquisition functions. A weakness of this approach is that it relies on random sampling to encourage exploration and therefore the performance of this method might suffer early on when the scale of the objectives is unknown or when either the input space or the objective space is high-dimensional [21, 64]. Another popular class of multi-objective acquisition functions are improvement-based. These strategies focus on improving a performance metric over sets, for example the hypervolume indicator [18, 19, 26, 93] or the R2 indicator [24]. The main drawback of these approaches is that the performance of these methods can be biased towards a single performance metric, which can be inadequate to assess the multi-objective aspects of the problem [98]. There are also many other multi-objective acquisition functions discussed in the litera-

36th Conference on Neural Information Processing Systems (NeurIPS 2022).

ture, which mainly differ by how they navigate the exploration-exploitation trade-off [8, 9, 52, 68, 69].

Instead of relying on scalarizations or an improvement-based criterion, this paper considers the perspective where the goal of interest is to improve the posterior distribution over the optimal points. We propose a novel information-theoretic acquisition function called the Joint Entropy Search (JES), which assesses how informative an observation will be in learning more about the joint distribution of the optimal inputs and outputs. This acquisition function combines the advantages of existing information-theoretic methods, which focus solely on improving the posterior of either the optimal inputs [31, 33, 39] or the optimal outputs [4, 6, 80]. We connect JES with the existing information-theoretic acquisition functions by showing that it is an upper bound to these utilities.

After acceptance of this work, we learnt of a parallel line of inquiry by Hvarfner et al. [46], who independently came up with the same JES acquisition function (3). Their work focussed on the single-objective setting and the approximation scheme they devised is subtly different to the ones we present. We see our work as being complementary to theirs because we both demonstrate the effectiveness of this new acquisition function in different settings.

**Contributions and organization.**   In Section 2, we set up the problem and introduce the novel JES acquisition function. In Section 3, we present a catalogue of conditional entropy estimates to approximate this utility and present a simple extension to the batch setting. These approximations are analytically tractable and differentiable, which means that we can take advantage of gradient-based optimization. The main results that we developed here can be viewed as direct extensions to the recent work in the Bayesian optimization literature by Suzuki et al. [80] and Moss et al. [59]. In Section 4, we present a discussion on the hypervolume indicator and explain how it can be a misleading performance criterion because it is sensitive to the scale of the objectives. We show that information-theoretic approaches are naturally invariant to reparameterization of the objectives, which make them well-suited for multi-objective black-box optimization. For a more complete picture of performance, we propose a novel weighted hypervolume strategy (Appendix K), which allows us to assess the performance of a multi-objective algorithm over different parts of the objective space. In Section 5, we demonstrate the effectiveness of JES on some synthetic and real-life multi-objective problems. Finally in Section 6, we provide some concluding remarks. Additional results and proofs are presented in the Appendix.

## 2   Preliminaries

We consider the problem of maximizing a vector-valued function $f : \mathbb{X} \to \mathbb{R}^M$ over a bounded space of inputs $\mathbb{X} \subset \mathbb{R}^D$. To define the maximum $\max_{\mathbf{x} \in \mathbb{X}} f(\mathbf{x})$, we appeal to the Pareto partial ordering in $\mathbb{R}^M$. For the rest of this paper, we will denote vectors by $\mathbf{y} = (y^{(1)}, \dots, y^{(M)}) \in \mathbb{R}^M$, the non-negative real numbers by $\mathbb{R}_{\geq 0}$ and diagonal matrices by $\mathrm{diag}(\cdot)$.

**Pareto domination.**   We say a vector $\mathbf{y} \in \mathbb{R}^M$ weakly Pareto dominates another vector $\mathbf{y}' \in \mathbb{R}^M$ if it performs just as well in all objectives if not better: $\mathbf{y} \succeq \mathbf{y}' \iff \mathbf{y} - \mathbf{y}' \in \mathbb{R}^M_{\geq 0}$. Additionally, if the vectors are not equivalent, $\mathbf{y} \neq \mathbf{y}'$, then we say strict Pareto domination holds: $\mathbf{y} \succ \mathbf{y}' \iff \mathbf{y} - \mathbf{y}' \in \mathbb{R}^M_{\geq 0} \setminus \{\mathbf{0}_M\}$, where $\mathbf{0}_M$ is the $M$-dimensional zero vector. This binary relation can be further extended to define domination among sets. Let $A, B \subset \mathbb{R}^M$ be sets, if the set $B$ lies in the weakly dominated region of $A$, namely $B \subseteq \mathbb{D}_{\preceq}(A) = \cup_{\mathbf{a} \in A}\{\mathbf{y} \in \mathbb{R}^M : \mathbf{y} \preceq \mathbf{a}\}$, then we say $A$ weakly dominates $B$, denoted by $A \succeq B$. In addition, if it also holds that the dominated regions are not equal, $\mathbb{D}_{\preceq}(A) \neq \mathbb{D}_{\preceq}(B)$, we say strict Pareto domination holds, denoted by $A \succ B$.

**Multi-objective optimization.**   The goal of multi-objective optimization is to identify the Pareto optimal set of inputs $\mathbb{X}^* = \arg\max_{\mathbf{x} \in \mathbb{X}} f(\mathbf{x}) \subseteq \mathbb{X}$. The Pareto set is defined as the set of inputs whose objective vectors are not strictly Pareto dominated by another: $\mathbf{x}^* \in \mathbb{X}^* \iff \mathbf{x}^* \in \mathbb{X}$ and $\nexists \mathbf{x} \in \mathbb{X}$ such that $f(\mathbf{x}) \succ f(\mathbf{x}^*)$. The image of the Pareto set in the objective space $\mathbb{Y}^* = f(\mathbb{X}^*) = \max_{\mathbf{x} \in \mathbb{X}} f(\mathbf{x})$ is called the Pareto front. For convenience of notation, we will denote the set of Pareto optimal input-output pairs by $(\mathbb{X}^*, \mathbb{Y}^*)$.

**Bayesian Optimization**   is a sample efficient global optimization strategy, which relies on a probabilistic model in order to decide which points to query. In Appendix A.1, we present the pseudo-code

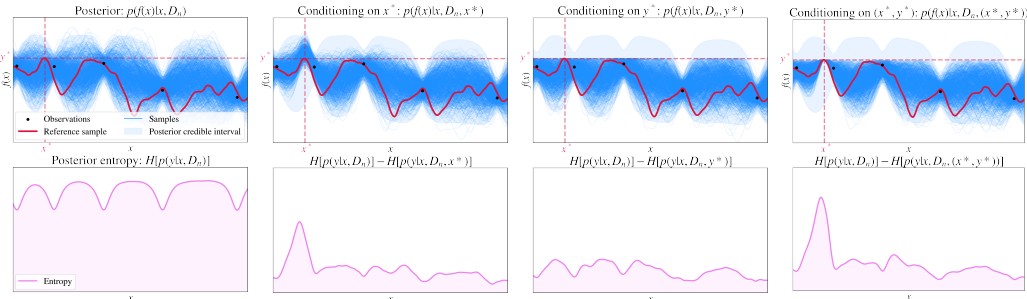

Figure 1: Comparison of the samples (top) and change in entropy (bottom) for the posterior and conditional distributions. The red line in the posterior plots denotes the reference sample that is used to obtain the maximizer $x^*$ and maximum $y^*$, whilst the shaded blue region is the $95\%$ credible interval of the posterior $p(f(x)|x, D_n)$. Conditioning on $x^*$ reduces the entropy for all inputs according to how correlated it is with $x^*$. Conditioning on $y^*$ reduces the entropy for all inputs according to the posterior probability that the objective surpasses $y^*$. Conditioning on $(x^*, y^*)$ leads to a drop in entropy based on both the input correlation with $x^*$ and the posterior probability of exceeding $y^*$.

for the standard BO procedure—for more details see [13, 29, 75]. In this work, we will use independent Gaussian process priors [71] on each objective, $f^{(m)} \sim \text{GP}(\mu_0^{(m)}, \Sigma_0^{(m)})$, where $\mu^{(m)} : \mathbb{X} \to \mathbb{R}$ is the mean function and $\Sigma^{(m)} : \mathbb{X} \times \mathbb{X} \to \mathbb{R}$ is the covariance function for objective $m$. The observations at location $\mathbf{x} \in \mathbb{X}$ will be assumed to be corrupted with additive Gaussian noise, $\mathbf{y} = f(\mathbf{x}) + \boldsymbol{\epsilon}$, where $\boldsymbol{\epsilon} \sim \mathcal{N}(0, \text{diag}(\boldsymbol{\sigma}(\mathbf{x})))$ denotes the observation noise with variance $\boldsymbol{\sigma}(\mathbf{x}) \in \mathbb{R}_{\geq 0}^M$. After $n$ evaluations, we have a data set $D_n = \{(\mathbf{x}_t, \mathbf{y}_t)\}_{t=1,\dots,n}$. The posterior model $p(f|D_n)$ is a collection of independent Gaussian processes $f^{(m)}|D_n \sim \text{GP}(\mu_n^{(m)}, \Sigma_n^{(m)})$. The explicit expressions for the mean and covariance are presented in Appendix A.2. The main focus of this work is on designing the acquisition function, $\alpha : \mathbb{X} \to \mathbb{R}$, which is used to select the inputs: $\mathbf{x}_{n+1} = \arg\max_{\mathbf{x} \in \mathbb{X}} \alpha(\mathbf{x}|D_n)$.

**Information-theoretic acquisition functions** focus on maximizing the gain in information from the next observation and a function of the probabilistic model. Initial work in BO focussed on picking points to learn more about the distribution of the maximizer $p(\mathbb{X}^*|D_n)$. Specifically, the goal of interest was to maximize the mutual information between the observation $\mathbf{y}$ and the Pareto set $\mathbb{X}^*$ conditional on the current data set $D_n$:

$$\alpha^{\text{PES}}(\mathbf{x}|D_n) = \text{MI}(\mathbf{y}; \mathbb{X}^*|\mathbf{x}, D_n) = H[p(\mathbf{y}|\mathbf{x}, D_n)] - \mathbb{E}_{p(\mathbb{X}^*|D_n)}[H[p(\mathbf{y}|\mathbf{x}, D_n, \mathbb{X}^*)]] \quad (1)$$

where $H[p(\mathbf{x})] = -\int p(\mathbf{x}) \log p(\mathbf{x}) d\mathbf{x}$ represents the differential entropy. This acquisition function is commonly referred to as predictive entropy search (PES) [39, 40, 74], but it was formerly[1] known as entropy search (ES) [38, 84]. Despite the importance of obtaining more information about the maximizer, the PES acquisition function is heavily dependent on the approximation of $p(\mathbf{y}|\mathbf{x}, D_n, \mathbb{X}^*)$, which is both computationally difficult to implement and optimize. This motivated researchers to consider a simpler scheme that focusses on learning more about the distribution of the maximum $p(\mathbb{Y}^*|D_n)$. The resulting acquisition function is known as the max-value entropy search (MES) [4, 44, 80, 86]:

$$\alpha^{\text{MES}}(\mathbf{x}|D_n) = \text{MI}(\mathbf{y}; \mathbb{Y}^*|\mathbf{x}, D_n) = H[p(\mathbf{y}|\mathbf{x}, D_n)] - \mathbb{E}_{p(\mathbb{Y}^*|D_n)}[H[p(\mathbf{y}|\mathbf{x}, D_n, \mathbb{Y}^*)]]. \quad (2)$$

Unlike PES, the conditional probability $p(\mathbf{y}|\mathbf{x}, D_n, \mathbb{Y}^*)$ arising in MES can be approximated and optimized more easily because some approximations lead to closed-form expressions. Despite the favourable properties of MES, the primary goal of interest is to identify the location of the maximizer $\mathbb{X}^*$ and not necessarily the value of the maximum $\mathbb{Y}^*$. To combine the advantages of both of these approaches, we propose the joint entropy search acquisition function, which focusses on learning more about the joint distribution of the optimal points $p((\mathbb{X}^*, \mathbb{Y}^*)|D_n)$:

$$\begin{aligned} \alpha^{\text{JES}}(\mathbf{x}|D_n) &= \text{MI}(\mathbf{y}; (\mathbb{X}^*, \mathbb{Y}^*)|\mathbf{x}, D_n) \\ &= H[p(\mathbf{y}|\mathbf{x}, D_n)] - \mathbb{E}_{p((\mathbb{X}^*, \mathbb{Y}^*)|D_n)}[H[p(\mathbf{y}|\mathbf{x}, D_n, (\mathbb{X}^*, \mathbb{Y}^*))]]. \end{aligned} \quad (3)$$

---

[1]The difference in the naming convention stems solely from the approximation strategy used to estimate the mutual information. At a high level, ES applies expectation propagation [58] to estimate $p(\mathbb{X}^*|D_n \cup \{\mathbf{x}, \mathbf{y}\})$, whilst PES applies expectation propagation to estimate $p(\mathbf{y}|\mathbf{x}, D_n, \mathbb{X}^*)$.

The JES acquisition function inherits the advantages of the PES and MES acquisition functions because it considers the knowledge learnt about the optimal points and is also simple to implement—more details in the next section. The following proposition shows that we can also interpret JES as an upper bound to both the PES and MES acquisition function.

**Proposition 1.** *The JES is an upper bound to any convex combination of the PES and MES acquisition functions:* $\alpha^{\text{JES}}(\mathbf{x}|D_n) \geq \beta\alpha^{\text{PES}}(\mathbf{x}|D_n) + (1-\beta)\alpha^{\text{MES}}(\mathbf{x}|D_n)$, for any $\beta \in [0,1]$.

In Figure 1, we illustrate the subtle differences between the different information-theoretic acquisition functions. More specifically, we visualise the difference between the conditional distributions arising in each acquisition function for a single-objective problem using one sample of the optimal points.

**Remark.** In the BO literature it is common to distinguish between single-objective and multi-objective acquisition functions by appending 'MO' to the end of the acronym. For notational simplicity, we opt against this convention in this paper. In Appendix C, we emphasize the main differences that arise when computing the information-theoretic algorithms in both settings.

## 3  Approximating JES

In this section, we present several approximations to the JES acquisition function (3) and a simple extension to the batch setting. The first term in the JES criterion (3) is the entropy of a multivariate normal distribution:

$$H[p(\mathbf{y}|\mathbf{x}, D_n)] = \frac{M}{2}\log(2\pi e) + \frac{1}{2}\sum_{m=1}^{M}\log(\Sigma_n^{(m)}(\mathbf{x}, \mathbf{x}) + \sigma^{(m)}(\mathbf{x})). \tag{4}$$

The second term is an intractable expectation which is approximated by drawing Monte Carlo samples from $p((\mathbb{X}^*, \mathbb{Y}^*)|D_n)$. The conditional entropy $H[p(\mathbf{y}|\mathbf{x}, D_n, (\mathbb{X}^*, \mathbb{Y}^*))]$ is also an intractable quantity which has to be estimated. The overall approximation of (3) will take the form

$$\hat{\alpha}^{\text{JES}}(\mathbf{x}|D_n) = H[p(\mathbf{y}|\mathbf{x}, D_n)] - \frac{1}{S}\sum_{s=1}^{S}h((\mathbb{X}_s^*, \mathbb{Y}_s^*); \mathbf{x}, D_n), \tag{5}$$

where $h$ denotes the conditional entropy estimate and $(\mathbb{X}_s^*, \mathbb{Y}_s^*) \sim p((\mathbb{X}^*, \mathbb{Y}^*)|D_n)$ are the Monte Carlo samples. The distribution $p(\mathbf{y}|\mathbf{x}, D_n, (\mathbb{X}^*, \mathbb{Y}^*))$ is very challenging to work with because it enforces the global optimality condition that the function lies below the Pareto front $f(\mathbb{X}) \preceq \mathbb{Y}^*$. Instead of enforcing global optimality, we make the common simplifying assumption as in [59, 80, 86] and only enforce the optimality condition at the considered location: $f(\mathbf{x}) \preceq \mathbb{Y}^*$. By applying Bayes' theorem, the resulting density of interest becomes

$$p(\mathbf{y}|\mathbf{x}, D_{n*}, f(\mathbf{x}) \preceq \mathbb{Y}^*) = \frac{p(f(\mathbf{x}) \preceq \mathbb{Y}^*|\mathbf{x}, D_{n+})}{p(f(\mathbf{x}) \preceq \mathbb{Y}^*|\mathbf{x}, D_{n*})}p(\mathbf{y}|\mathbf{x}, D_{n*}), \tag{6}$$

where we have denoted the augmented data sets by $D_{n*} = D_n \cup (\mathbb{X}^*, \mathbb{Y}^*)$ and $D_{n+} = D_{n*} \cup \{(\mathbf{x}, \mathbf{y})\}$. We will refer to the quantity $p(f(\mathbf{x}) \preceq \mathbb{Y}^*)$ as the cumulative distribution function (CDF). The following lemma shows that this CDF can be computed analytically when the set $\mathbb{Y}^* \subset \mathbb{R}^M$ is discrete. This is a standard result [16, 50, 68, 80] which can be derived by first partitioning the region of integration, $\mathbb{D}_{\preceq}(\mathbb{Y}^*) = \cup_{\mathbf{y}^* \in \mathbb{Y}^*}\{\mathbf{z} \in \mathbb{R}^M : \mathbf{z} \preceq \mathbf{y}^*\}$, into a collection of hyperrectangle subsets and then summing up the individual contributions—see Figure 2 for a visual. This partition can be computed using an incremental approach (Algorithm 1 of [55]), which has a cost of $O(|\mathbb{Y}^*|^{\lfloor M/2 \rfloor + 1})$. In the single-objective setting, the maximum is a single point $y^* \in \mathbb{R}$ and the box-decomposition is simply the interval $\mathbb{D}_{\preceq}(\{y^*\}) = (-\infty, y^*]$.

**Lemma 1.** *Let* $\mathbb{Y}^* \subset \mathbb{R}^M$ *be a finite set and* $\mathbf{z} \sim N(\mathbf{a}, diag(\mathbf{b}))$ *be an $M$-dimensional multivariate normal with mean* $\mathbf{a} \in \mathbb{R}^M$ *and variances* $\mathbf{b} \in \mathbb{R}_{\geq 0}^M$. *Let* $\mathbb{D}_{\preceq}(\mathbb{Y}^*) = \bigcup_{j=1}^{J}B_j = \bigcup_{j=1}^{J}\prod_{m=1}^{M}(l_j^{(m)}, u_j^{(m)}]$ *be the box decomposition of the dominated space, then*

$$p(\mathbf{z} \preceq \mathbb{Y}^*) = \sum_{j=1}^{J}\prod_{m=1}^{M}\left[\Phi\left(\frac{u_j^{(m)} - a^{(m)}}{\sqrt{b^{(m)}}}\right) - \Phi\left(\frac{l_j^{(m)} - a^{(m)}}{\sqrt{b^{(m)}}}\right)\right]. \tag{7}$$

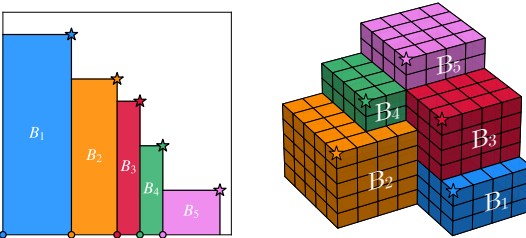

Figure 2: Box decompositions for a two-dimensional and three-dimensional Pareto front.

In Algorithm 1, we present the pseudo-code for the estimation of the JES acquisition function at a single candidate input. Several variables that calculated within the algorithm are independent of the input (coloured in blue). For computational efficiency, we only compute these variables once and then save them to memory for later use.

---

**Algorithm 1:** Joint Entropy Search (JES).

**Input:** A candidate $\mathbf{x}$; the data set $D_n$.
  // Cached variables are coloured in blue.
1 Compute the initial entropy $h_0 = H[p(\mathbf{y}|\mathbf{x}, D_n)]$.
2 **for** $s = 1, \ldots, S$ **do**
3      Sample a path $f_s \sim p(f|D_n)$.
4      Compute the Pareto optimal points $\mathbb{X}_s^* = \arg\max_{\mathbf{x}' \in \mathbb{X}} f_s(\mathbf{x}')$ and $\mathbb{Y}_s^* = f_s(\mathbb{X}_s^*)$.
5      Compute the box decomposition $\mathbb{D}_{\preceq}(\mathbb{Y}_s^*) = \bigcup_{j=1}^J B_j$.
6      Compute the conditional $p(f|D_n \cup (\mathbb{X}_s^*, \mathbb{Y}_s^*))$.
7      Compute the estimate $h_s = h((\mathbb{X}_s^*, \mathbb{Y}_s^*); \mathbf{x}, D_n)$.
8 **end**
9 **return** $\hat{\alpha}^{\text{JES}}(\mathbf{x}|D_n) = h_0 - \frac{1}{S}\sum_{s=1}^S h_s$.

---

### 3.1 Estimating the conditional entropy

The entropy of (6) can be written as an $M$-dimensional expectation over the multivariate normal distribution $p(\mathbf{y}|\mathbf{x}, D_{n*}) = \mathcal{N}(\mathbf{y}; \boldsymbol{\mu}_{n*}(\mathbf{x}), \boldsymbol{\Sigma}_{n*}(\mathbf{x}, \mathbf{x}))$:

$$
\begin{aligned}
&H[p(\mathbf{y}|\mathbf{x}, D_{n*}, f(\mathbf{x}) \preceq \mathbb{Y}^*)] \\
&= -\mathbb{E}_{p(\mathbf{y}|\mathbf{x}, D_{n*})}\left[\frac{p(f(\mathbf{x}) \preceq \mathbb{Y}^*|\mathbf{x}, D_{n+})}{p(f(\mathbf{x}) \preceq \mathbb{Y}^*|\mathbf{x}, D_{n*})}\log\left(\frac{p(f(\mathbf{x}) \preceq \mathbb{Y}^*|\mathbf{x}, D_{n+})}{p(f(\mathbf{x}) \preceq \mathbb{Y}^*|\mathbf{x}, D_{n*})}p(\mathbf{y}|\mathbf{x}, D_{n*})\right)\right].
\end{aligned} \tag{8}
$$

To simplify the notation, we define the $m$-th standardized value by

$$
\gamma_m(z) = (z - \mu_{n*}^{(m)}(\mathbf{x}))/\sqrt{\Sigma_{n*}^{(m)}(\mathbf{x}, \mathbf{x})} \tag{9}
$$

for any scalar $z \in \mathbb{R}$. Using this function together with Lemma 1, we denote the cumulative distribution $p(f(\mathbf{x}) \preceq \mathbb{Y}^*|\mathbf{x}, D_{n*})$ by $W = \sum_{j=1}^J W_j = \sum_{j=1}^J \prod_{m=1}^M W_{j,m}$, where

$$
W_{j,m} = \Phi(\gamma_m(u_j^{(m)})) - \Phi(\gamma_m(l_j^{(m)})) \tag{10}
$$

are the differences appearing in (7). Moreover, we denote the differences of the first derivative of $W_{j,m}$ and the negative of the second derivative (with respect to $\gamma_m$) by

$$
G_{j,m} = \phi(\gamma_m(u_j^{(m)})) - \phi(\gamma_m(l_j^{(m)})), \tag{11}
$$

$$
V_{j,m} = \gamma_m(u_j^{(m)})\phi(\gamma_m(u_j^{(m)})) - \gamma_m(l_j^{(m)})\phi(\gamma_m(l_j^{(m)})), \tag{12}
$$

where $\phi$ is the probability density function of a standard normal distribution. In the setting where the observation noise is zero, the conditional distribution is a truncated multivariate normal, which is known to have an analytical equation formula for the entropy (Theorem 3.1. in [80]). In Appendix E,

we construct an ad hoc extension to this expression when the observation noise is non-zero.

In the noisy setting, the distribution of interest is a type of multivariate skew normal distribution, which is known to not have an analytical form for the entropy [1]. As a result, we propose two approximation strategies to estimate this entropy. The first strategy is to approximate the integral using Monte Carlo. The details of the Monte Carlo estimate $h^{\text{JES-MC}}$ is described in Appendix F. The second strategy is to directly approximate the distribution with one that exhibits an analytical entropy. We consider the most obvious choice, which is a multivariate normal distribution with the same first two moments. The same strategy was proposed in [59] for the single-objective multi-fidelity MES acquisition function. By a standard result (Chapter 12 of [17]), the entropy of this approximating distribution is actually an upper bound for the entropy of interest: $H[p(\mathbf{y}|\mathbf{x}, D_{n*}, f(\mathbf{x}) \preceq \mathbb{Y}^*)] \leq \frac{M}{2} \log(2\pi e) + \frac{1}{2} \log \det \mathbb{V}\text{ar}(\mathbf{y}|\mathbf{x}, D_{n*}, f(\mathbf{x}) \preceq \mathbb{Y}^*)$. The following result shows that the these central moments can be computed analytically.

**Proposition 2.** *Under the modelling set-up outlined in Section 2, for an input $\mathbf{x} \in \mathbb{X}$ the first and second central moment of $p(\mathbf{y}|\mathbf{x}, D_{n*}, f(\mathbf{x}) \preceq \mathbb{Y}^*)$ are*

$$\mathbb{E}[y^{(m)}|\mathbf{x}, D_{n*}, f(\mathbf{x}) \preceq \mathbb{Y}^*] = \mu_{n*}^{(m)}(\mathbf{x}) - \frac{\sqrt{\Sigma_{n*}^{(m)}(\mathbf{x}, \mathbf{x})}}{W} \sum_{j=1}^{J} W_j \frac{G_{j,m}}{W_{j,m}}$$

*and*

$$\text{Cov}\left(y^{(m)}, y^{(m')} \middle| \mathbf{x}, D_{n*}, f(\mathbf{x}) \preceq \mathbb{Y}^*\right)$$

$$= \begin{cases} \frac{\sqrt{\Sigma_{n*}^{(m)}(\mathbf{x},\mathbf{x})}\sqrt{\Sigma_{n*}^{(m')}(\mathbf{x},\mathbf{x})}}{W} \sum_{j=1}^{J} W_j \frac{G_{j,m}}{W_{j,m}} \left(\frac{G_{j,m'}}{W_{j,m'}} - \frac{1}{W} \sum_{j'=1}^{J} W_{j'} \frac{G_{j',m'}}{W_{j',m'}}\right), & m \neq m'; \\ \Sigma_{n*}^{(m)}(\mathbf{x}, \mathbf{x}) + \sigma^{(m)}(\mathbf{x}) - \frac{\Sigma_{n*}^{(m)}(\mathbf{x},\mathbf{x})}{W} \left(\sum_{j=1}^{J} W_j \frac{V_{j,m}}{W_{j,m}} + \frac{1}{W} \left(\sum_{j=1}^{J} W_j \frac{G_{j,m}}{W_{j,m}}\right)^2\right), & m = m'. \end{cases}$$

As an upper bound on the conditional entropy leads to a lower bound on the mutual information, we will refer to the resulting conditional entropy estimate as the JES-LB estimate:

$$h^{\text{JES-LB}}((\mathbb{X}^*, \mathbb{Y}^*); \mathbf{x}, D_n) = \frac{M}{2} \log(2\pi e) + \frac{1}{2} \log \det \mathbb{V}\text{ar}(\mathbf{y}|\mathbf{x}, D_{n*}, f(\mathbf{x}) \preceq \mathbb{Y}^*). \quad (13)$$

We could obtain a further lower bound by ignoring the off-diagonal terms in the covariance matrix. We dub the resulting approximation as the JES-LB2 entropy estimate:

$$h^{\text{JES-LB2}}((\mathbb{X}^*, \mathbb{Y}^*); \mathbf{x}, D_n) = \frac{M}{2} \log(2\pi e) + \frac{1}{2} \sum_{m=1}^{M} \log \mathbb{V}\text{ar}(y^{(m)}|\mathbf{x}, D_{n*}, f(\mathbf{x}) \preceq \mathbb{Y}^*). \quad (14)$$

Figure 3 presents an illustration of the different density approximations that are used within the various conditional entropy estimates. An important remark is that all the conditional entropy estimates that we have developed here can also be applied to estimate MES. The only difference in the MES algorithm is that we no longer apply the conditioning step (line 6 of Algorithm 1) because we are interested in estimating $H[p(\mathbf{y}|\mathbf{x}, D_n, f(\mathbf{x}) \preceq \mathbb{Y}^*)]$ as opposed to (8). Consequently, the MES acquisition function is cheaper to evaluate because the cost of evaluating the posterior variance at a single input is $O(n^2)$, whereas JES incurs a cost of $O((n + |\mathbb{Y}^*|)^2)$—more details are presented in the cost analysis in Appendix H.

## 3.2 Batch evaluations

Evaluating the JES acquisition functions for a batch of points $\mathbf{x}^{[1:q]} = (\mathbf{x}^{[1]}, \ldots, \mathbf{x}^{[q]}) \in \mathbb{X}^q$ is expensive because the entropy estimates now depends on the $q$-dimensional normal CDF and its derivatives. To circumvent this issue, we follow the example of [59] and propose a suboptimal batch approach by upper bounding the expensive joint conditional entropy term by the sum of the individual entropies: $H[p(\mathbf{y}^{[1:q]}|\mathbf{x}^{[1:q]}, D_{n*}, f(\mathbb{X}) \preceq \mathbb{Y}^*)] \leq \sum_{i=1}^{q} H[p(\mathbf{y}^{[i]}|\mathbf{x}^{[i]}, D_{n*}, f(\mathbb{X}) \preceq \mathbb{Y}^*)]$. The resulting $q$-batch lower bound JES estimate is given by

$$\hat{\alpha}^{q\text{LB-JES}}(\mathbf{x}^{[1:q]}|D_n) = H[p(\mathbf{y}^{[1:q]}|\mathbf{x}, D_n)] - \frac{1}{S} \sum_{s=1}^{S} \sum_{i=1}^{q} h((\mathbb{X}_s^*, \mathbb{Y}_s^*); \mathbf{x}^{[i]}, D_n), \quad (15)$$

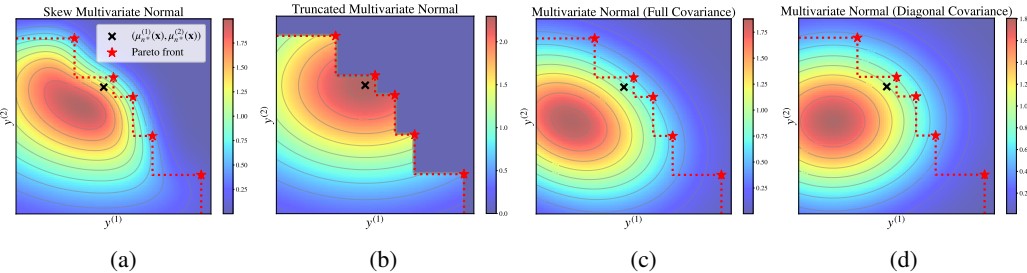

Figure 3: Comparison of the density approximations to the skew multivariate normal distribution $p(\mathbf{y}|\mathbf{x}, D_{n*}, f(\mathbf{x}) \preceq \mathbb{Y}^*)$ shown in (a) for a single input $\mathbf{x} \in \mathbb{X}$ and sample Pareto front $\mathbb{Y}^*$. A zero noise assumption leads to the truncated multivariate normal approximation shown (b), whilst a moment matching approach leads to the multivariate normal approximations in (c) and (d).

where $h$ is the conditional entropy estimate and

$$H[p(\mathbf{y}^{[1:q]}|\mathbf{x}, D_n)] = \frac{M}{2}\log(2\pi e) + \frac{1}{2}\sum_{m=1}^{M}\log\det(\Sigma_n^{(m)}(\mathbf{x}^{[1:q]}, \mathbf{x}^{[1:q]}) + \mathrm{diag}(\sigma^{(m)}(\mathbf{x}^{[1:q]}))) \quad (16)$$

is the initial entropy. This acquisition function is defined over a $qD$-dimensional space, which becomes more difficult to optimize as $q$ increases. Alternatively, we can maximize this function greedily by sequentially selecting the best input conditioned on the previously chosen points. This greedy procedure turns out to be an effective strategy when the acquisition function is submodular [88]. In Appendix G, we show that this batch acquisition function is indeed submodular.

## 4   Performance criteria

In multi-objective optimization, the most common way to measure performance is by comparing the approximate Pareto set $\hat{\mathbb{X}}^*$ against the optimal Pareto set $\mathbb{X}^*$ in the objective space: $d(f(\hat{\mathbb{X}}^*), f(\mathbb{X}^*))$ where $d : 2^{\mathbb{R}^M} \times 2^{\mathbb{R}^M} \to \mathbb{R}$ is a function that measures the discrepancy between the sets of objective vectors. Existing work in multi-objective BO mainly focusses on the hypervolume (HV) discrepancy, $d_{\mathrm{HV}}(A, B) = |U_{\mathrm{HV}}(A) - U_{\mathrm{HV}}(B)|$, where the HV indicator, $U_{\mathrm{HV}}(A) = \int_{\mathbb{R}^M} \mathbb{I}[\mathbf{r} \preceq \mathbf{z} \preceq A]d\mathbf{z}$, is defined as the volume between a reference point $\mathbf{r} \in \mathbb{R}^M$ and a set $A \subset \mathbb{R}^M$. The general guidance is to set reference point to be slightly worse than the nadir, which is the vector consisting of the worst possible points, $\min_{\mathbf{x} \in \mathbb{X}} f^{(m)}(\mathbf{x})$, for objectives $m = 1, \dots, M$—see [47] for more details.

An attractive feature of the HV indicator is that it is Pareto complete (or compliant) in the sense that a better set will lead to a larger HV [98]: $A \succ B \implies U_{\mathrm{HV}}(A) > U_{\mathrm{HV}}(B)$, if we assume the sets $A$ and $B$ are finite. The reverse implication known as Pareto compatibility does not hold for the HV indicator [98]. In other words, the HV can be used to discriminate between sets where one dominates another, but it cannot be relied upon when the sets are incomparable. Not all incomparable sets are treated equally by the HV indicator [96]. For instance Figure 4a shows an example where the HV indicator places more emphasis on the end points of the Pareto front. On other hand, if we apply a monotonically increasing transformation $g_m : \mathbb{R} \to \mathbb{R}$ to each objective, the Pareto set will not change, whereas the HV comparison will (Figure 4b). Implicitly, the HV indicator assumes that a linear change in one objective is equivalent to a linear change in another. This assumption might not necessarily reflect the decision maker's outlook and this is something that is typically overlooked when designing and benchmarking multi-objective optimization algorithms. The following result shows that information-theoretic acquisitions function are in fact agnostic to the choice of parameterization.

**Proposition 3.** *The information-theoretic acquisition functions $\alpha^{\mathrm{PES}}$, $\alpha^{\mathrm{MES}}$ and $\alpha^{\mathrm{JES}}$ are invariant to reparameterization of the objective space that are consistent with the Pareto ordering relations. For example, $\alpha^{\mathrm{JES}}(\mathbf{x}|D_n) = \mathrm{MI}(\mathbf{y}; (\mathbb{X}^*, \mathbb{Y}^*)|D_n) = \mathrm{MI}(g(\mathbf{y}); (\mathbb{X}^*, g(\mathbb{Y}^*))|D_n)$, where the $g_m : \mathbb{R} \to \mathbb{R}$ is a strictly monotonically increasing function acting only on the $m$-th objective.*

To benchmark the algorithms, we use both the standard HV discrepancy (Section 5) and the HV discrepancy under different parameterizations (Appendix L). To easily obtain a family of parameteri-

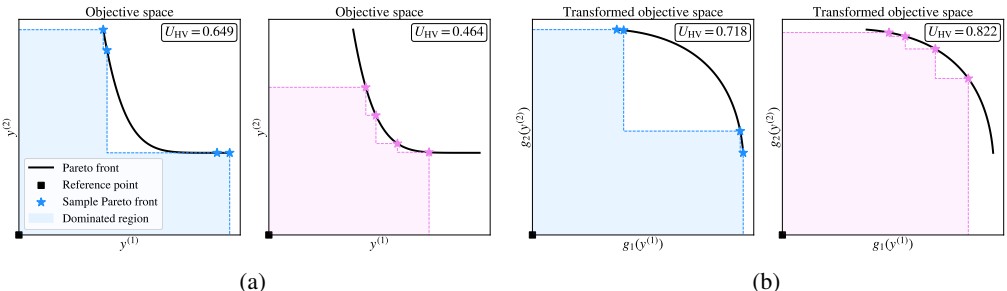

Figure 4: Comparison of the HV of two sample Pareto fronts in (a) the standard objective space $\mathbf{y} = (y^{(1)}, y^{(2)})$ and (b) the transformed objective space $g(\mathbf{y}) = (g_1(y^{(1)}), g_2(y^{(2)}))$ described in Appendix K. The HV indicator prefers a different set depending on the choice of parameterization.

zations, we devise a novel weighting approach in Appendix K, which exploits the fact that the HV indicator can be written as an expectation over a uniform distribution on the $(M-1)$-dimensional hypercube [23, 94]. We observe that it is possible to assess the performance at different locations of the objective space by using alternate distributions over the hypercube. We call the resulting metric the generalized hypervolume (GHV). In our experiments, we found that the performance of each algorithm changed with regards to the choice of parameterization, but the JES approaches tended to be one of the strongest performers throughout.

## 5 Experiments

We empirically evaluate the JES acquisition function on a range of synthetic and real-world benchmark problems. We compare this approach with some popular acquisition functions in multi-objective BO: TSEMO [12], ParEGO [51], NParEGO [19], EHVI [18], NEHVI [19], PES [31, 33] and MES-0 [80]. We have also included the MES-LB, MES-LB2 and MES-MC acquisition functions, which can be easily derived from the conditional entropy estimates that we developed here. All algorithms are based on the open source Python library BoTorch [3], which uses features from GPyTorch [30] for Gaussian process regression and PyTorch [66] for automatic differentiation. All experiments are repeated using 100 different initial seeds and we generate the Pareto set recommendation $\hat{\mathbb{X}}^*$ of 50 points by maximizing the posterior mean using a multi-objective solver (NSGA2 [22] from the Pymoo library [10]). The complete details of the experiments are outlined in Appendix L, whilst the code is available at https://github.com/benmltu/JES.

### 5.1 Benchmarks

**Synthetic benchmark.**   We consider the ZDT2 [22] benchmark with $D = 6$ inputs and $M = 2$ objectives. We corrupt the observations with additive Gaussian noise with zero-mean and standard deviation set to approximately $10\%$ of the objective ranges.

**Chemical reaction.**   This benchmark considers a nucleophilic aromatic substitution reaction (SnAr) between 2,4-difluoronitrobenzene and pyrrolidine in ethanol to produce a mixture of a desired product and two side-products [45]. The design space comprises of $D = 4$ components relating to the initial conditions. The goal is to optimize $M = 2$ objectives, namely the space time yield and the environmental impact. We apply a logarithm transform to the objectives and contaminate the observations with additive Gaussian noise with zero-mean and standard deviation set to approximately $3\%$ of the resulting objective ranges in order to emulate a potential real-world scenario.

**Pharmaceutical manufacturing.**   This problem is concerned with optimizing the Penicillin production process outlined in [56]. The design space is made up of $D = 7$ elements that control the initial condition of the reactions. The goal is to optimize $M = 3$ objectives, which relates to the yield, the amount of carbon dioxide released and the time to ferment. We include additive zero-mean Gaussian noise with a standard deviation set to approximately $1\%$ of the objective ranges.

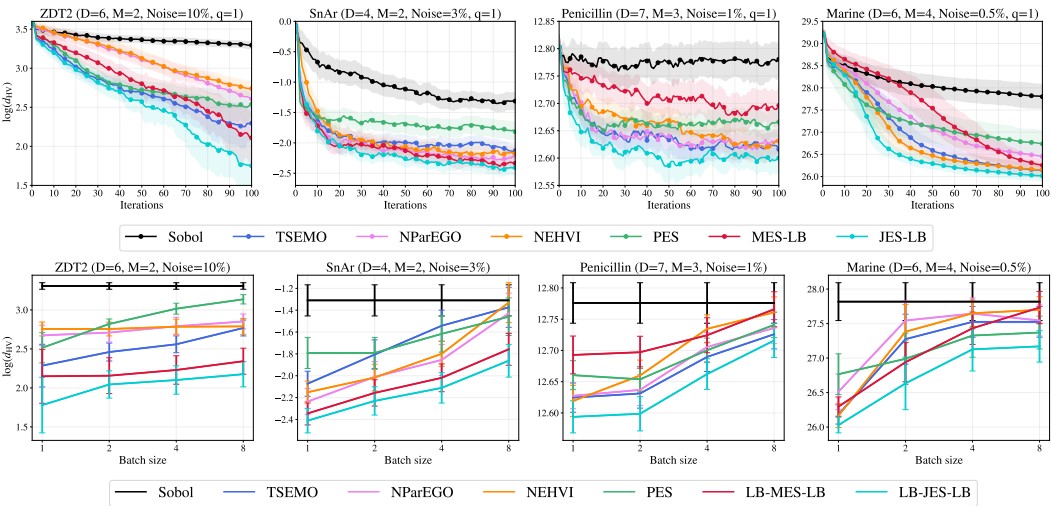

Figure 5: A comparison of the mean logarithm HV discrepancy with two standard errors over one hundred runs for the four benchmark problems on a subset of the algorithms. We present the sequential and batch results on the top and bottom, respectively.

**Marine design.** This problem considers optimizing a family of bulk carriers subject to the constraints imposed for ships travelling through the Panama Canal [65, 73]. The design space is made up of $D = 6$ variables that determine the architecture of the carriers. The goal of this problem is to maximize the annual cargo, whilst minimizing the transportation cost and the ship weight subject to some design constraints. We consider the reformulation in [83], which converts the constraints into another objective. For this reformulated $M = 4$ objective problem, we corrupt the observations with additive zero-mean Gaussian noise with standard deviation set to approximately $0.5\%$ of the objective ranges.

## 5.2 Results and discussion

We present the log HV discrepancy results for both the sequential and batch experiments in Figure 5. The JES approach is consistently one of the stronger performing algorithms for these set of experiments. A similar conclusion is reached when we consider the weighted variant of the hypervolume in Appendix L.8.

**Conditional entropy estimates.** We compared the performance of the different conditional entropy estimates for both the JES and MES acquisition function in Appendix L.6. We observed that in the majority of cases all the estimates exhibit similar performance. As a result, we recommend using the cheapest approximation, which is usually the lower bound estimates, judging from the wall times presented in Appendix L.9.

**Acquisition wall times.** The wall times in Appendix L.9 indicate that the cost of acquiring a new point with JES is comparable with NEHVI, slightly more expensive than MES, but cheaper than PES. We note that the wall times for all methods can be improved by taking advantage of parallelization. In particular, for entropy based methods we used a gradient-free optimizer to sequentially optimize the multi-objective samples (line 4 in Algorithm 1), whereas in practice we should of ideally solved these problems in parallel using a gradient-based optimizer such as [57].

**Querying high performing points.** In certain domains it might be useful to directly query high-performing points because the final decision will be restricted to only the sampled locations $X_N$. In Appendix L.5, we investigated the performance when such a restriction was made. In this setting, we observed that the information-theoretic approaches were occasionally outperformed by the improvement and scalarization based acquisition functions, which picked points more greedily. We observed that the entropy based approaches had a tendency to pick points that are more informative for the posterior over the optimal points as opposed to directly selecting a point that is known to

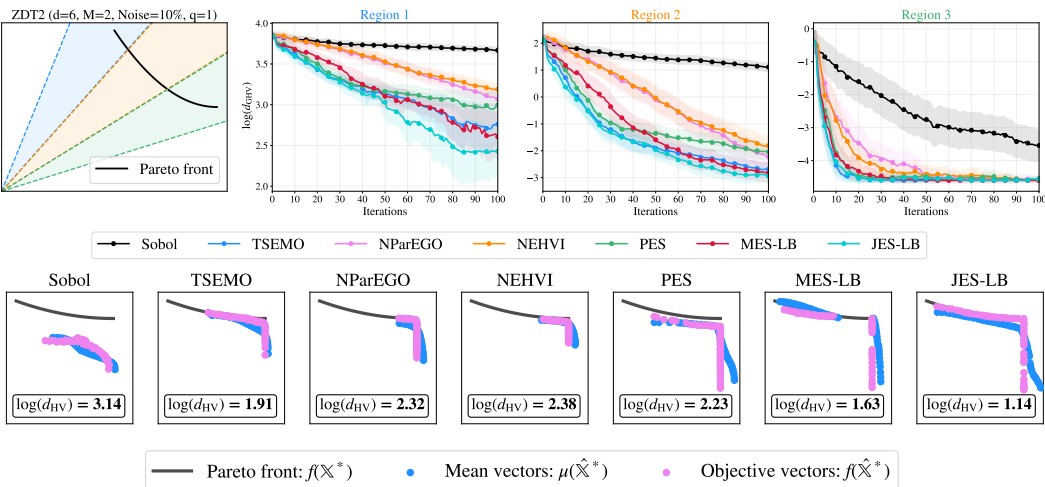

Figure 6: An example of the generalized hypervolume on the ZDT2 benchmark. On the top plot we present the mean logarithm GHV discrepancy results for three different regions. On the bottom plot, we present the Pareto front and its approximation at the final time for the run which achieved the top 20th percentile on the standard hypervolume.

perform well. To address this setting, we recommend combining information-theoretic acquisition functions with an epsilon greedy approach, where points are occasionally picked according to a greedy strategy such as maximizing a function of the posterior mean.

**Assessing local performance.** Using the generalized hypervolume, we can target different parts of the objective space in order to get a much better picture of performance. We demonstrate this on a simple bi-objective example in Figure 6, where we assess that quality of the approximations at three different regions of the objective space. We observe that all of the BO algorithms were quickly able to identify the right section of the Pareto front. Evidently, the main source of difficulty for this problem arises from approximating the points in the left section of the Pareto front, which favours the second objective. This observation would not be apparent if we focussed solely on the standard HV.

**General guidance.** The ideal acquisition functions is problem dependent and strongly depends on the decision maker's plans and goals. In a completely black-box setting, where there is no immediate preferences, Proposition 3 and the empirical results motivates the usage of information-theoretic acquisition functions, which treats all points on the Pareto front as equally desirable a priori.

## 6 Conclusion

We introduced JES, a novel information-theoretic acquisition function for multi-objective BO. To approximate this acquisition function, we presented several approximations to the conditional entropy and also a simple extension for the batch setting. Experimental results suggest that JES is very competitive with existing acquisition functions in terms of the HV discrepancy and its weighted variants. The main limitation of the JES acquisition function is that it relies on routines such as box decompositions and multi-objective optimization of function samples, which can be expensive to execute for very large problems. Future work could focus on improving the scalability of these information-theoretic methods and extending it to more general settings, which include constrained, decoupled and multi-fidelity optimization—see Appendix M for more details.

## Acknowledgments and Disclosure of Funding

BT was supported by the EPSRC StatML CDT programme EP/S023151/1 and BASF SE, Ludwigshafen am Rhein. NK was partially funded by JPMorgan Chase & Co. under J.P. Morgan A.I. Faculty Research Awards 2021.

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
