# OpenReview forum: "Joint Entropy Search for Multi-Objective Bayesian Optimization"
_NeurIPS.cc/2022/Conference — NeurIPS 2022 Accept_

### Official Review · Reviewer_LjBd · 2022-07-10

**Rating:** 7
**Confidence:** 5
**Soundness:** 3 good
**Presentation:** 3 good
**Contribution:** 3 good

**Summary:**

This paper proposes a novel information-theoretic acquisition function for BO called Joint Entropy Search (JES), which considers the joint information gain for the optimal set of inputs and outputs. The authors present several analytical approximations to the JES acquisition function and also introduce an extension to the batch setting. The authors show the effectiveness of this new approach on a range of synthetic and real-world problems in terms of the hypervolume and its weighted variants

**Questions:**

No

**Limitations:**

 the authors adequately addressed the limitations of their work. i.e., the computational complexity of JES.

**Strengths And Weaknesses:**

Strength 1, it identifies the problem of predictive entropy search (PES) method, i.e., the acquisition function is heavily dependent on the approximation of p(y|x, Dn, X∗),  which is difficult to implement and optimize. Besides, it also shows that the max-value entropy search (MES) method, though comes with closed-form expressions, does not directly dealing with the maximum Y∗. JES combines the advantages of both methods.

Strength 2, it gives several theoretic property of the JES function, with in-depth study.

Strength 3, the experiments are very detail, and have a good overall design.

---

> ### Author Response · Authors · 2022-08-01
> **To LjBd**
>
> Many thanks for the positive feedback.

---

### Official Review · Reviewer_WzLK · 2022-07-11

**Rating:** 7
**Confidence:** 4
**Soundness:** 3 good
**Presentation:** 4 excellent
**Contribution:** 3 good

**Summary:**

The paper proposes a new information-based acquisition function for multi-objective Bayesian optimization that combines the approaches from the known PES and MES acquisition functions. The authors derive various approximations to efficiently compute this acquisition function and provide a comparison of their performance characteristics against a number of baselines.


**Questions:**

- l34: " The main drawback of these approaches is that the performance of these methods are biased towards a single scalar utility criterion, which can be inadequate to assess the multi-objective aspects of the problem". I don't really understand this point; after all, the scalar utility is defined specifically in order to capture the multi-objective aspects. In a sense your formulation is no different in that ultimately you optimize a scalar function, the information gain. Can you please clarify?
- l38: This sentence makes it seem as if other papers do not consider a "probabilistic perspective", which of course they do (this is BO after all).
- l156: what does "cancelling out" mean here? JES is computed on the covariance using the augmented set of observations so there shouldn't be any cancelling in the strict sense. Please clarify.

Nits / Typos:
- l28/29: it enables -> they enable
- l47: set-up -> set up
- l89: "and a property of the probabilistic model." <- not sure what is meant by this, please clarify


**Limitations:**

Yes.

**Strengths And Weaknesses:**

# Strengths:
- The paper is well written, in particular Section 2 provides a nice buildup of background to understand JES in relation to (P)ES and MES.
- The motivation for studying the new acquisition function and the theoretical contributions (in terms of JES and the various approximations) are solid; I view them primarily as well crafted technical work on top of a large body of existing literature in the field rather than a surprising or particularly novel set of contributions (which is not a bad thing).
- The main strength of the paper beyond this solid execution is that it provides a comprehensive comparison and ablation studies of various state-of-the art acquisition functions.
- The authors also provide a full implementation of the methods in a popular and maintained python library for Bayesian Optimization. This in itself is a significant contribution to the community, which can now easily use not just JES but also the other implemented baselines (without trying to get an unmaintained library such as spearmint running in a modern software environment...).
- Pointing out the potential issues with HV w.r.t. rescaling objectives is valuable, and the fact that JES is invariant to that is nice and useful (though not surprising as one would expect that from an information-based acquisition function).
- The supplementary material is very comprehensive and goes into lots of detail. This is very useful to the reader. For instance, I found the illustrations of the acquisition function in the contour plots instructive.

# Weaknesses:
- One shortcoming of the work is this that there aren't any variance estimates of the performance of the different algorithms in the empirical evaluation. The checklist states that "We do not include the error bars directly on the plots because it made it difficult to visually compare between the different algorithms.", but as a reader I would have hoped to see a more thorough analysis of this in the supplementary material. The quantile versions of the plots are useful to some extent, and by comparing the performance of the suggested methods across the detailed results of the benchmarks the reader can get a sense that the proposed algorithms may indeed improve over existing methods. However, this is a lot of work to require from the reader, and I would ask the authors to add a more comprehensive discussion to help practitioners understand in which the proposed methods are most useful and computationally feasible.
- The information gain estimates rely crucially on the assumption that the different objective functions are modeled by independent GPs. If the objectives are correlated, then this assumption could result in poor performance. Other approaches, such as Monte-Carlo based EHVI methods are not subject to these limitations since their methodology applies in the same fashion to multi-output GPs. In fact, as they are based on MC sampling from the model's posterior, such methods are also compatible with other probabilistic models, which is not the case for the approach proposed in this paper.
- Overall, I found the experimental section of the paper in the main text too brief to be able to get a good sense for the empirical performance of the JES approach in comparison to the baselines (both in terms of sample efficiency and computational footprint). I would suggest the authors move some of the mathy and notation-heavy details of Section 3  (e.g. Proposition 3) to the appendix, potentially streamline the exposition in other parts of the paper, and include additional results / discussion of the empirical performance in the main text.

---

> ### Author Response · Authors · 2022-08-01
> **To WzLK**
>
> - W1 (error bars): We agree that this was an omission from our part, we have now restructured the plots and included error bars in all the figures. We also included some more discussion in the experiments section to guide practitioners.
>
> - W2 (independent models): Regarding independence and GP modelling, this is a valid point and we accept this criticism. Our main focus here was on improving the acquisition and hence chose this standard setting. As the reviewer points out, further work is needed to adapt our methodology as well as other competing methods based on GPs for the correlated case.
>
> - W3 (discussion): We also agree with the reviewer that more details were needed in the experiment section. We have now moved a proposition to the Appendix, which allowed us to augment and improve the discussion of the main results and findings.
>
> - Q1 (scalar utility): Multi-objective optimization algorithms are usually assessed using performance metrics defined over the space of subsets of vectors. One acquisition strategy is to directly optimize the performance metric directly e.g. the hypervolume. An algorithm that does this will likely perform well under this specific performance metric, but may not appear to do well under another performance metric, which might assess other features of the Pareto front. We believe that working to improve the quality of a posterior distribution is more convenient and sidesteps these issues. We also tried to include different measures of performance to emphasize this. We agree that this statement was confusing and have now rephrased this accordingly.
>
> - Q2 (probabilistic perspective): We agree and have removed the phrase.
>
> - Q3 (cancellation): This is a fair point. It is correct the cancellation only occurs when computing the MES-0 estimate. As discussed in another review, this is an ad hoc modification in order to maintain the exploration-exploitation trade-off. We have now relegated this result and discussion to the Appendix D as it caused confusion and drew attention away from the main results.
>
> - Q4 (property of the model): We meant to say any function of the model f, for example the argmax(f) or max(f) in the case of PES and MES. We have now rephrased this.
>
> Typos:
>
> Thanks for spotting the typos, they have now been sorted.

---

> > ### Comment · Reviewer_WzLK · 2022-08-05
> > **Thank you for your response**
> >
> > Thanks for the response and for making the editorial changes and adding the variance to the plots. Unfortunately, in a lot of these examples it's hard to discern the performance of at least some of the methods given the relatively high variance. It would be good to consider evaluations that make the performance differences more clear. I never considered the performance of the method a particularly compelling aspect of the paper, but the paper does have other sufficiently many other strengths and so I am sticking with my score.

---

### Official Review · Reviewer_Z8pJ · 2022-07-11

**Rating:** 6
**Confidence:** 3
**Soundness:** 3 good
**Presentation:** 3 good
**Contribution:** 3 good

**Summary:**

A new acquisition function for Bayesian Optimization (JES) is proposed in this paper. It considers the joint information gain for the optimal set of X and Y.  Several analytical approximations to the acquisition function are also introduced.

**Questions:**

- It is not quite clear from the paper why this new method is superior to the existing methods (for example PES and MES as mentioned in the paper) with respect to its cost-performance trade-off. The amount of work involved in using this new acquisition function may be large, especially compared to MES.
- Maybe it is good to identify certain specific settings or scenarios where this joint information gain acquisition function has an advantage over both PES and MES.
- For motivation and illustration purposes, maybe in the introduction, some small examples on a small instance can be worked out where this new method can be easily compared with PES and MES. In this way, this method can be better motivated.
- The approximation methods can achieve different accuracy levels using a different number of samples, if possible, some discussion or guidance on that will be helpful in that section.

**Ethics Review Area:**

["I don’t know"]

**Limitations:**

Not applicable.

**Strengths And Weaknesses:**

Strength: A new information-theoretic acquisition function for Bayesian Optimization is presented, and approximation methods are also discussed and analyzed in this paper.

Weakness: It is not quite clear why this new method is superior to the existing methods (for example PES and MES as mentioned in the paper) with respect to its cost-performance trade-off.

---

> ### Author Response · Authors · 2022-08-01
> **To Z8pJ**
>
> - W1 and Q1a (why JES): In general, the best acquisition function is dependent on both the problem and the decision maker's goal. We recommend using JES when the goal is to obtain a good posterior distribution over the optimal points. The experiments indicate that JES is indeed a good choice when that is the goal.
>
> - Q1b (JES vs MES): Our main result concerning the lower bound conditional entropy estimate (Prop. 2) benefits both the MES and JES acquisition function. The cost of JES is indeed slightly greater than the MES because of the conditioning step in Algorithm 1, but in turn leads to a strategy that queries points which are more informative for both the optimal inputs and outputs.
>
> - Q1c (cost performance trade-off): Investigating the cost-performance trade-off in multi-objective Bayesian optimization is certainly an interesting research question that we are also looking in to. As Section 4 suggest, it is unclear how to define performance from the get-go and this choice will naturally have an effect on any analysis we conduct to compare between cost and performance.
>
>
> - Q2 (example): Our initial investigation into this resulted in the contour plots in Appendix H. We observed that JES strikes a different balance on the exploration-exploitation trade-off compared to the PES and MES. For a single-objective problem, where the posterior mean is bi-modal, the JES opts for one mode whilst the PES and MES opts for the other. For this particular set up, the mode that JES picks is marginally better.
>
> - Q3 (motivation): This is a very good suggestion and is something we tried to accomplish as we discussed above. In the end, we opted for Figure 1, which we believe is sufficient as an initial way to motivate the difference between the acquisition functions.
>
> - Q4 (approximation): We observed very little difference when increasing the number of samples in Appendix J.3. Our results indicate that setting the number of Monte Carlo samples moderately to 10 seems to work well. We observed that increasing the number of Pareto optimal points tended to result in an improvement in performance (up to a certain point) at the expense of computational cost, we also set this to 10 as a compromise. Developing theoretical results in order to justify these choices, such as Theorem 4.1 in Takeno et al. (2021), is something we are also looking into.

---

> > ### Comment · Reviewer_Z8pJ · 2022-08-07
> > **Thanks for the response**
> >
> > Thanks for the detailed response from the authors! There seems to be some work needed to be done for the comparison and making a more compelling case for this method, so I am sticking with the score.

---

### Official Review · Reviewer_QtD3 · 2022-07-12

**Rating:** 4
**Confidence:** 5
**Soundness:** 3 good
**Presentation:** 4 excellent
**Contribution:** 2 fair

**Summary:**

This work proposes a novel acquisition function JES for multi-objective Bayesian optimization based on the joint entropy of the optimal inputs and outputs. Previously works have studied input and output entropy independently. The work proposes novel entropy approximations—particularly one based on moment-matching, which has been previously used in the single objective setting but not the multi-objective setting. The work extends the proposed JES acquisition function to the batch setting using an additional approximation. In addition, this work analyzes potential for negative pathologies when using the hypervolume indicator for evaluation and proposes an alternative weighted hypervolume evaluation criterion.

**Questions:**

* How well does the constrained version work empirically?
* How robust is the performance to the noise level?
* The JES approach seems quite incremental.
  * The input entropy and output entropy are well-studied
  * The moment-matching approximation is well-studied in the single objective case [50]
  * The box decomposition-based entropy computation is proposed in [68]
  * Lemma 1 is not novel. It comes directly from [68]  and is used in [68] which should be made more prominent
  * The whole JES computation algorithm is an incarnation of BAX with a different acquisition function than PES and a different approximation method.
* JES does not support modeling correlated outcomes---independent GPs models are assumed to be used. Why then distinguish equations (10) and (11)? Also, Why does Figure 3 show correlation between outcomes? Aren’t the covariances matrices across outcomes for a single point always diagonal under the assumption of the independent GP models?
* The batch extension follows [50] and is “suboptimal” (L193) and approximates the joint JES across a batch of points with the sum of the independent entropies of each point. Why would the best batch of `q` points not be the same point `q` times using this acquisition function?
  * Also, sequential greedy batch selection is common practice. While the submodularity proof is neat, it applies to 1) approximating true batch JES with sequential greedy or 2) approximating the coarse approximation (sum of independent entropy) of batch JES with sequential greedy. It makes no guarantees about how well the coarse approximation (12) approximates batch JES (or some approximate version of JES without the batch independent entropy approximation (12)). This error seems like it could be quite large.
  * How does JES scale with increasing batch size both in terms of wall time and in terms of optimization performance? It is hard to draw conclusions from q=1 and q=8 alone.
* Why does this paper focus on multi-objective JES and not a single objective JES?
* Being invariant to component-wise monotonic transformations of the objectives is a nice property of entropy-based methods (prop 4). But this is not specific to JES, and therefore seems a bit odd to include.
* Page 7 footnote: “The reference point is typically chosen to be the nadir…”. Typically the reference point is typically chosen to be *slightly worse* than the nadir of the *Pareto frontier*, *not the entire objective space* as stated in the footnote. Using the nadir of entire objective space (as described) would often be a poor choice of reference point because it would be very far from the pareto frontier. Was this strategy used for evaluation purposes (hypervolume computation) in the experiments? Also using the nadir of the Pareto frontier directly would be a poor choice because it gives zero hypervolume to the end points, which is why the reference point is typically chosen to be slightly worse than the nadir (see Ishibuchi et al., 2011).
  * L954/983: using a nadir over all observed points as the reference point is also is likely a poor choice for solving the MOOP problem for recommendation and for use in NEHVI and EHVI. A point that is slightly worse than the nadir point over the pareto frontier across the observed designs is likely a much better choice because it 1) will ensure that it is not very far from the current pareto frontier, which could be the case with using the nadir across all observed points and 2) gives non-zero hypervolume to the endpoints of the pareto frontier and beyond the endpoints, which is likely very important for discovering designs outside of the currently observed Pareto frontier with EHVI/NEHVI.
  * It is very hard to distinguish the trend lines in Figure 5. I would recommend focusing on one JES method. This would also help the reader understand which JES method is best and should be used. Currently, this is very hard to parse. The colors for TSEMO and LB-JES-0 are very similar.
  * Also, can you provide error bars please? Currently, it is not clear which differences in performance are statistically significant
* How does the performance look on noiseless problems?
* L288: Regarding being able to specify a preferred region (via the reference point) with hypervolume-based methods, this seems like a positive. It seems like a major drawback of entropy-based methods to only be able to identify the entire Pareto frontier and not just an area of interest.
* For problems such as the ZDT family, the optimal designs lie in a small region of design space. This means that the heuristic for selecting the initial points for gradient-based acquisition optimization is particularly important for improvement-based acquisition functions since they can be 0 over a large portion of the design space. In L903, the paper says that initial points were not sampled near the Pareto frontier, which would have led to faster convergence for entropy-methods. I think it would be even more important for improvement-based methods. Can you provide empirical results were the initialization heuristic does sample points near the current best (pareto optimal) for acquisition optimization?

Minor comments

L24: “trade-off” -> “trade-offs”

L31: “might suffer when the scale of the objectives is unknown apriori”. A common approach is simply to rescale the objectives to the unit cube based on the observed outcomes

L32: “or when the input space is high-dimensional”. Wouldn’t high-dimensional output spaces be more problematic for random scalarizations?

L50: “analytically available, which enables the use of gradient-based optimization”. It is not clear here why analytic availability implies differentiable

L73: should be \subseteq instead of \subset

L196: These acronyms are very confusing

L298: it would be good to clarify that “multi-objective optimization” refers to the fact that computing JES requires solving a set of multi-objective optimization problems of optimizing GP function samples.


**Limitations:**

Some limitations are discussed in the last section, but further discussion around when to use which method would be useful. Additionally, characterizing drawbacks of the approach and sensitivity of the approach and entropy-based methods more generally to model-specification would be useful. Negative societal impacts are not discussed.

**Strengths And Weaknesses:**

Although this paper is largely well-written and provides precise detail on the empirical setup and evaluation, the contributions are limited. Multi-objective BO is an important but well-studied space. Hence, the work can, inherently, at best be incremental. Furthermore, entropy-based methods have been studied in multi-objective BO. The contributions of a novel joint input/output entropy acquisition function and a new multi-objective entropy approximation show improvement over previous ES methods, but these are small contributions that leverage well-studied ideas from the literature. Although JES approximations show good empirical performance, the evaluation is limited to evaluating the out-of-sample model predicted Pareto frontier. This evaluation technique favors and is well-suited to information theoretic methods, but as the paper states often time decision-makers would prefer to launch a previously evaluated design (regardless of whether this is observation noise). Therefore, it is not clear that JES approximations outperform alternative approaches such as DGEMO and NEHVI when the decision maker is only interested in choosing a final design at implementation time from the set of previously evaluated designs. Evaluation of the performance using this in-sample Pareto frontier (under the noiseless objectives) would help provide more clarity. Furthermore, the absense of error bars in the empirical evaluation makes it impossible to determine which differences in performance are statistically significant. Moreover, the paper does not propose one single JES approximation to use. There are numerous JES methods and many confusing acronyms, and the empirical evaluation does not obviously indicate a single best JES method (even based on the mean performance, without accounting for uncertainty). In addition, JES is computationally intensive and is slower than even EHVI-based methods---which are known to be computationally demanding.

The additional contributions of an analysis of the HV indicator and the proposed weighted HV metric are a bit orthogonal to the new JES method and feel out of place with the rest of the paper, which focuses on JES. It seems like the weighted HV metric should be de-emphasized in the main text (specifically, the contributions), since it is not actually used in the main text and it is unclear what distribution is appropriate for sampling scalarization weights. The weighted HV and HV analysis are not JES specific, but is a general comment on multi-objective optimization that is out-of-place with the JES-focus of the paper.

It is worth noting that code is provided and well-documented. This is particularly notable because PES is known to be difficult to implement and most BO papers leverage an old Spearmint implementation with now unsupported dependencies. A high-quality PES implementation is of value to the community.

On a whole, a) the contributions are scattered (results on JES, results on ES methods handling monotonic transformations, an analysis of using HV for evaluating multi-objective optimization algorithms), b) the JES-specific contributions are incremental (a new joint entropy and a new application of an approximation techniques from the single objective setting), and c) the empirical evaluation is difficult to interpret (which JES method is best/should be used, what does performance look like if only in-sample---rather than out-of-sample---designs can be used in the Pareto frontier) and lacks measures of statistical significance.

---

> ### Author Response · Authors · 2022-08-01
> **To QtD3**
>
> - W1 (in sample results): In the revision, we have included the results for the in-sample experiments in Appendix J.4. We have also included a discussion in the main text.  The in-sample results reveal that the information-theoretic algorithms have tendency not to directly select the best performing points and instead opt to query more informative points that have higher uncertainty. This can result in a behaviour where the in-sample performance appears to stagnate, whilst the out-of-sample performance improves. We mention that if directly querying high performing points is useful, taking an epsilon greedy approach where one occasionally picks points by maximizing a greedy criterion might be a useful modification.
>
> - W2 (error bars): We have now restructured all the plots and have included error bars.
>
> - W3 (computation cost): We have been quite thorough about testing the wall times of our different conditional entropy approximations and acquisition functions using a single processor (Appendix J.3 and J.7). Our approaches, MES-LB and JES-LB results in competitive wall times with other competing methods even without using parallelization, which could improve the run time further.
>
> - W4 (many approximations): We have indeed proposed multiple implementations for JES. Our intention was to be methodical and thorough on how JES can be implemented and made more efficient. The main observation is that the JES approximations can perform similarly for the same number of Monte Carlo samples and sampled Pareto optimal points. The best choice can be selected based on wall-times. Our experiments showed the benefit of using the lower bound estimates for both the multi-objective MES and JES when only using CPU computations. Some routines could parallelized for further speed gains. For example, the vector optimization (step 4 in Algorithm 1) was executed serially using a gradient-free optimization method. Potentially this can be solved in parallel using gradient-based optimization such as the one proposed by Liu et al. (2021).
>
> - W5 (weighting the hypervolume): Our impression is that there is over-emphasis in the recent literature on maximization of the hypervolume. This can be misleading in some cases, so the aim of Section 4 and Figure 4 is to highlight this and propose a more complete assessment and comparison. Clearly our intention is to focus more on information theoretic methods (and hence include Prop. 3 on invariance to reparameterization) but we believe that the point of working towards more diverse comparisons is valuable for all methods. Interestingly our probabilistic weighting idea can be viewed as an extension of Zitzler et al (2007), who identified this issue from their early work (Zitzler et al 1998) that first introduced the hypervolume criterion.
>
> As regards to the overall assessment, we respectfully disagree with a) and b) but understand the reviewer's point of view and thank them for the feedback. Regarding point c) on empirical evaluation, we believe our revisions has addressed this.
>
> - Q1 (constraints): Early numerical results with constraints are encouraging, but more work is needed to report concrete conclusions.
>
> - Q2 (noise levels): In the paper we used different noise levels in the different examples, but in numerical work not presented here we did try different noise levels for each example and noticed fairly similar results.
>
> - Q3a (incremental and references): Input and output entropy are well studied, but until this paper have not been used jointly. We believe is a natural but not straightforward extension. We have now further emphasized the connection with earlier work in the main work. In particular, we have added additional references to Moss et al. (2021) and Suzuki et al. (2020) throughout the paper and especially in the contributions section. Moreover, we have emphasized that Lemma 1 and the box-decomposition strategy are standard results in multi-objective optimization dating back to earlier work by Keane (2012), Couckuyt et al. (2014), Picheny (2015) and Suzuki et al. (2020).
>
> - Q3b (BAX): The BAX acquisition function (Neiswanger et al 2021) is subtly different to the one considered here. The probability density of observations in the BAX paper (Equation 5 in Neiswanger et al 2021) conditions only on the "execution path", which in our setting is the augmented data set. In contrast, the probability density of the observations in our work (Equation 6) conditions on both the "execution path" and the optimality condition $f(x) \preceq Y^*$. We have now included this point in the related work section and cited Neiswanger et al (2021).
>
> - Q4 (independent models): We accept the criticism that the different objectives are modelled as independent. The probability density in Figure 3 conditions on the points, data, and the optimality condition. The latter causes the correlations and this can also be seen in the results in Prop. 2.

---

> > ### Author Response · Authors · 2022-08-01
> > **To QtD3**
> >
> > - Q5 (batch lower bound): In general, the optimizer of a lower bound might not be an optimizer of the original quantity. Batch size experiments can be very expensive and the lower bound approximations are necessary from a practical point of view. We agree that the estimation error and role of batch size should be investigated further. This is part of current work that look into this in more detail and use also ideas from stochastic control and dynamic programming.
> >
> > - Q6 (single-objective): We found the vector optimization setting to be more interesting and challenging because we also have to consider the multi-objective trade-off. In preliminary investigations to single-objective optimization, we noticed little difference between the many different acquisition functions for single-objective BO. The acquisition functions mainly differ in how they balance the exploration-exploitation trade-off. This was the motivation of the contour plots in Appendix H.
> >
> > - Q7 (invariance): In addition to introducing JES a general aim in the paper is to highlight how information theoretic criteria can be improved and what benefits they provide compared to other approaches. As such we believe it is worth including here.
> >
> > - Q10a (nadir points): This point is correct and we clarified this in the revised main text. The reference point in the evaluation for the experiments did indeed use a reference point that is slightly worse than the nadir for the whole function (not just the Pareto front).
> >
> > - Q10b (error bars): Thank you for the suggestion, we have now restructured the plots to focus solely on the lower bound estimate and have modified the figures to include error bars.
> >
> > - Q11 (noiseless): Noiseless results were similar to what is presented in the paper and hence we did not include them.
> >
> > - Q12 (incorporating preferences): We would not go as far as suggesting that this is drawback for entropy based methods (or any method), simply because not all methods are designed to do this. If this is of interest all methods should be modified and extended for such a scenario. For example, to extend entropy based methods we could redefine the optimal front as one which is Pareto optimal and dominates a specified reference point. Computationally, this modification will result in different bounds for the box-decomposition for MES and JES.
> >
> > - Q13 (warm start): We have not tried this form of initialisation near the Pareto frontier. This is an interesting suggestion and we believe this could improve the efficiency of all methods. Better understanding how to effectively solve the inner optimization problem in BO is an interesting line of inquiry which we leave for future work. Some interesting references we are aware of include Gramacy et al. (2021) and Grosnit et al. (2021) - Are we Forgetting about Compositional Optimisers in Bayesian Optimisation.
> >
> > Minor points:
> >
> > We agree with most points and have taken onboard the reviewer suggestions. Regarding input or output dimension, the answer is both are challenging for random scalarization. In the work by Paria et al (2020), they proved a cumulative Bayes regret for an acquisition strategy based random scalarization, which depended on both the input dimension and number of objectives. We have rephrased accordingly. Regarding gradients as the reviewer hints it is the form the expression rather than analytical tractability per se. We have also rephrased this accordingly.

---

> > > ### Comment · Reviewer_QtD3 · 2022-08-05
> > > **response**
> > >
> > > Thanks to the authors for the thorough response!
> > >
> > > > W1 (in sample results):
> > >
> > >  Thanks for adding these
> > >
> > > > W2 (error bars):
> > >
> > > Thanks for adding these. Unfortunately, it shows that in general on the test problems considered, JES is not statistically significantly better than other methods---even with 100 replicates and on a log scale. Furthermore, since not clearly faster than alternative approaches that are competitive w.r.t. to hypervolume, it is not clear why one should use JES
> > >
> > > > W3 (computation cost): Our approaches, MES-LB and JES-LB results in competitive wall times with other competing methods even without using parallelization, which could improve the run time further.
> > >
> > > It is worth noting that NEHVI for example is much faster on a GPU than a CPU (and measurements in this paper are made on a single CPU_, so while runtime of MES-LB/JES-LB might be improved by exploiting parallelism, the same is true of alternative approaches.
> > >
> > > >  W4 (many approximations):
> > >
> > > Thanks for reducing the number of methods in the plots. They are much clearer.
> > >
> > > > W5 (weighting the hypervolume):
> > >
> > > I agree that alternative evaluation metrics are interesting to consider, but including such a discussion in the main text strikes me as out of place---especially considering that no alternative metric is used for evaluation in the main text.
> > >
> > > > Q4 (independent models):
> > >
> > > Makes sense. Figure 3 seems out of place of place to me though because it is never even discussed in the paper
> > >
> > > > Q5 (batch lower bound): … We agree that the estimation error and role of batch size should be investigated further.
> > >
> > >  Thanks for confirming this. Unfortunately, this reaffirms my concerns about the critical potential failure modes of the batch approximation.
> > >
> > > I have increased the confidence of my review, but I will leave my score as is. The lack of statistically significant improvement in any test problem *even on a log scale with 100 replicates* makes it clear that the method does not offer a significant improvement in optimization performance. Since it isn't faster than alternatives, the question is when should this method be used. Furthermore, it is surprising to jump to the multi-objective setting with JES rather than the single objective setting---"In preliminary investigations to single-objective optimization, we noticed little difference between the many different acquisition functions for single-objective BO."---given that there are not significant differences in the multi-objective setting.
> > >
> > > The response to my critique about the suboptimal batch formulation confirmed my concern that the estimation error can be quite large and is not captured in their sub-modularity argument, which makes the batch formulation and sub-modularity argument not very confidence inspiring.

---

> > > > ### Comment · Area_Chair_Luur · 2022-08-08
> > > > **Author response**
> > > >
> > > > Hi authors,
> > > >
> > > > QtD3 and WzLK raise some important points with respect to statistical significance of the results with respect to alternative methods.  Your work is notable in its extensive evaluation of high-quality implementations of entropy-based methods for MOBO.  However, some of the evaluation is less transparent due to seemingly arbitrary choices of noise and parallaelism.  Plots that are devised to more directly quantify how well the AFs can handle increasingly higher levels of parallelism or noise are important for understanding the tradeoffs between different methods.  For example, in the qNEHVI paper, the authors consider final log HV regret as a function of batch size (Fig 4) and noise (Fig 14).  Something similar could be helpful here.  It is not necessary for JES (or the other ES) methods to strictly dominate any other methods, but it is important to investigate and clearly state when there are and are not practically significant differences between methods.  How do you plan to address these concerns?

---

> > > > > ### Author Response · Authors · 2022-08-08
> > > > > **Response**
> > > > >
> > > > > Thank you for the replies.
> > > > >
> > > > > - R1 (batch acquisition function): The lower bound batch acquisition function and its approximation are principled acquisition functions because they can be written as determinantal point processes (DPPs) (Kulesza et al 2012). This property was derived and used in the proof of submodularity. As discussed in the BO literature (references below), batch acquisition functions based on DPPs are powerful because they can promote diverse batches in high-quality regions. The trade-off between diversity and quality (Section 3.1 in Kulesza et al 2012) is evident in the form of the DPP kernel, Km_{i,j} = q_i q_j A_{i, j} (line 821), where q_i = exp(zeta_i/M) can be interpreted as the quality of item i, whilst A_{i,j} corresponds to a notion of similiarity. The Monte Carlo error arises in estimating the expectation in zeta_i (equation 31) and it contributes only to the relative quality assigned to each point in the batch. The optimal batch for the approximation might differ slightly due to the error present in the quality term, but the diversity of the batch will still be high because of the matrix A. The sequential greedy optimization of this acquisition function is arguably necessary because maximizing the original expression is known to be NP-hard (Ko et al. 1995).
> > > > >
> > > > >   - Ko et al. (1995) - An exact algorithm for maximum entropy sampling.
> > > > >
> > > > >   - Contal et al. (2013) - Parallel Gaussian Process Optimization with Upper Confidence Bound and Pure Exploration
> > > > >
> > > > >   - Kathuria et al. (2016) - Batched Gaussian Process Bandit Optimization via Determinantal Point Processes
> > > > >
> > > > >   - Wang et al. (2017) - Batched High-dimensional Bayesian Optimization via Structural Kernel Learning
> > > > >
> > > > > - R2 (more experiments): We are conducting more experiments similar to the ones listed in order to demonstrate the performance trade-off with different amount of noise and batch sizes. As the wall-time plots might suggest, our ability to generate these results within the rebuttal period are limited.

---

> ### Comment · Reviewer_QtD3 · 2022-08-09
> **Follow-up**
>
> My recommendations to the authors are:
> 1) Make clear up front about the sources of error in the batch acquisition function: the potential of error from the lower bound (this error relative to true batch acquisition function seems possibly quite large), MC error and submodularity error (the latter two really only apply to approximating the lower bound).
> 2) Draw a clearer boundary between the previous work and the new contributions. Currently, a careful read is required to distinguish between the two in the manuscript's current form.
> 3) Move the weighted hypervolume indicator to the appendix or include d_GHV results in the main text. Currently, the weighted hypervolume discussion seems out of place.
> 4) I second the AC that more comprehensive noise and batch experiments are warranted.

---

### Official Review · Reviewer_ucDe · 2022-07-17

**Rating:** 7
**Confidence:** 5
**Soundness:** 3 good
**Presentation:** 1 poor
**Contribution:** 3 good

**Summary:**

The paper proposes a new information gain acquisition function for multi-objective Bayesian optimization. The method selects the input that maximizes the joint information gain about both input and output space. The paper provides several approximations of the acquisition function using lower bound and assumptions about the observed variance. The method is extended to batch selection accompanied by a submodularity study. The paper provides an analysis of the sensitivity of the hypervolume performance metric to functions transformations.

**Questions:**

Approximations
+ How is the 0 variance assumption made, and what are the consequences of this on the selections made by the acquisition function? Canceling the effect of the predictive variances can lead to more exploitation in the acquisition function. It seems to be a little ad hoc just to add an additional term to cancel the variance effect. The transition is not explained well or justified.

Writing and presentation: suggestions for readability enhancement:
+ Several equations and inequalities are written in the text without any numbering and then used in the derivation without a referral.
+ I suggest adding all the equalities/inequalities and definitions of variables used in separately numbered equations. I am aware of the space constraint, but several definitions can fit in one line, and white space reduction can also be used.
+ Adding the propositions/lemmas written in the main paper near their corresponding proofs in the appendix would make it easier to follow the derivations. The reader won’t need to jump back and forth between the main paper and the appendix.
+ The connections and substitutions used in the derivations should be detailed, at least in the appendix.
+ f(x) <= Y* is a notation used repeatedly in the paper. However, the relation between two Pareto fronts is not always quantified as higher or equal and would depend on the tradeoff between several functions. A more rigorous notation should be used.
+ The confusion between SO and MO surfaces in the names used for algorithms and in the notation used in the paper. For example, in algorithm 1, line 3 can be confused as a single objective sampling or sampling from a single multi-task GP while each of the objectives is sampled separately.

Experiments:
+ Only batch sizes of 1 and 8 are used. Typically batch BO papers include extensive experiments on different batch sizes
+ More synthetic experiments can be useful to show the behavior of the algorithm when the number of functions and number of dimensions is varied
+ The noise choice for each experiment is different. How is this choice made?
+ Figure 13 does not easily provide the needed information. It would be more informative to plot all the algorithms using the same number of samples in the same figure and have different figures for different numbers of samples
+ Some relevant and important baselines were not cited or compared to:  Diversity-Guided Multi-Objective Bayesian Optimization With Batch Evaluations, Neurips 2020
+ The standard errors are not plotted. The use of quantiles is unconventional and do not substitute or provide the same interpretation as the error. A light-colored error plot can still be readable.
+  The performance improvement is minor, and the baselines sometimes work better.
+ In the Penicillin experiment, the hypervolume progress seems unstable for all algorithms; what is causing the fluctuation, and how is this interpreted?
+ Figure 1 provides an analysis based on single objective optimization, which does not necessarily hold in the multi-objective case. This figure is not discussed or analyzed anywhere in the text. The same issue applies to Figures 6, 7, and 10. I suggest removing these studies. They are not relevant, and they add to the confusion between SO and MO.
+ The discussion on the sensitivity of the hypervolume as a metric is insightful. However, it is important to note that it is a common practice to fit the Gaussian processes to normalized data, and therefore the scale issues between the objectives become less problematic. Additionally, several other distance-based metrics overcome some of the weaknesses of HV when comparing equivalent Pareto fronts. Therefore, evaluating the performance using more than one metric is encouraged.

Typos:
+ Typo in line 186 “because the cost of evaluating”
+ The structure of the sentence in line 176,177 is incorrect


**Limitations:**

The paper discussed some of the work's limitations.

**Strengths And Weaknesses:**

Strengths:
+ The paper is technically rich and provides an extensive study of the problem space with several approximations and derivations
+ The idea of the use of joint information gain is novel and interesting.
+ The extension to batch setting seems naïve however, the discussion of submodularity of the acquisition function is interesting
+ The paper provides a thorough cost analysis in terms of analytical complexity and wall clock time of optimization

Weaknesses:
+ Some design choices made in the approximation are not justified and seem ad hoc, including the cancellation of variance effect in the first approximation and choice of noise in experiments.
+ Many parts of the derivation in the main paper and the appendix are inspired by [50] and [68]. However, this does not seem to be properly acknowledged in the paper.
+ The paper has a significant presentation issue. In general, the derivations are very hard to follow, both in the main paper and the appendix. This is not due to the extensive mathematical formulation of the acquisition function but rather to the lack of smoothness in the writing and the assumption that the reader can easily follow all the equations and substitutions. I provide more details in the questions box.
+ The experimental setup has several weaknesses that I discuss more in detail in the questions and comments box
+ The writing of the paper assumes the reader is familiar with previous work on Information-theoretic acquisition functions, which is not necessarily true. Several steps of the algorithm are not described thoroughly and have confusing notations.
+ The paper uses “incorrect” acronyms to refer to previous work. This is problematic because the acronyms used in this paper are usually used to refer to different algorithms in SOO. For example, MES and PES are usually used to refer to a single objective acquisition function. In MO, they are referred to as PESMO and MESMO. The EHVI is usually used to refer to the sequential EHI method. The method discussed in this paper is the recently developed batch approach qEHVI and qNEHVI.

---

> ### Author Response · Authors · 2022-08-01
> **To ucDe**
>
> - W1 (some ad hoc design choices): This is indeed an ad hoc approximation and is meant to provide more intuition. This should have been better clarified in the text to avoid any confusion. We moved this result to Appendix D and added a further discussion, which we will briefly summarise here. In the zero observation noise setting, the JES-0 acquisition function can be written as $\mathbb{E}[-\log(W)]$ + other terms, where as a reminder $W = p(y \preceq Y^* | D_{n*})$  i.e. the probability that y is dominated by the sample Pareto front Y* conditioned on the augmented data set. The expectation here is over the distribution of optimal inputs and outputs. The first term $\mathbb{E}[-\log(W)]$  can be interpreted as the exploitation term because it computes the expectation of the negative log-probability of being dominated by the front. The other terms can be interpreted as an exploration term. The ad hoc correction can be viewed as a way to maintain the right amount of exploration even in the presence of noise. This correction demonstrates decent numerical results that are in-line with the other approximations. The contour plots in Appendix G demonstrates that this approximation is close for a single-objective example where the noise in this experiments is set to around 5\%.
>
> - W2 (reference to prior work): This is true, our analytical computations makes use and extends the results in these works. We have now further emphasized this in the main text and have also included mention in the contributions section.
>
> - W3 (presentation): Based on feedback from the reviewer we have updated the proofs with background notation and numbering where necessary. We have included the statement of the result before the proofs to improve the reader experience. We also used color-coded references and links throughout the document for ease of navigation.
>
> - W4 (experiments): Please see below.
>
> - W5 (familiarity with information-theoretic acquisitions): We agree that there is limited discussion on information theoretic acquisition functions, but at the same time this is a very well studied topic in Machine Learning as well as traditional Statistics and experimental design. For the notation, we tried to compromise between being close to previous works, readability and smooth exposition. For the algorithm we present standard mathematical pseudo-code and will make Python code available.
>
> - W6 (acronyms use): We agree that we deviated from using PESMO, MESMO etc and opted for notation such as PES, MES for simplicity. We have now added a mention at the end of section 2 to alert the reader on this.
>
> Approximations
>
> - Q1 (dealing with 0 variance assumptions): This has been discussed in W1 above.
>
> Writing and presentation
>
> - Q1-4: We have followed most suggestions from the reviewer - thanks for the feedback. In the Appendix we have added the statements proposition and lemmas before the proof and repeated the necessary equations with numbers.
>
> - Q5 ($f(x) \preceq Y^*$ notation): We have included a definition on the partial ordering relation on sets in the paragraph on Pareto domination in Section 2.
>
> - Q6 (pseudo-code): The algorithm is valid for both the single-objective and multi-objective setting as $f$ can be a random vector-valued function with M components. This is registered at the beginning of Section 2.
>
> Experiments
>
> - Q1 (batch sizes): This is a fair point. We note that the batch case was meant as an extension and had we had more time would have tested additional batch sizes.
>
> - Q2 (experiments): We believe we have a variety of cases with different dimensions and number of objective. More synthetic examples would indeed be useful but we preferred to focus more on standard benchmarks and more challenging problems.
>
> - Q3 (noise choice for different examples): The choice was based on what noise level we felt could potentially be more realistic in practice based on the context of the application. In numerical work not presented we did experiment with different noise levels and the results were rather similar.
>
> - Q4 (on Figure 13): The point here is all methods perform very similarly to the hypervolume criterion even when the number of Monte Carlo samples S varies and believe both ways of plotting will achieve this. Here we were more interested to check how each method performs with S separately. Since all plots are in the same scale so we believe that comparisons between methods are also possible here but differences are marginal.

---

> > ### Author Response · Authors · 2022-08-01
> > **To ucDe**
> >
> > - Q5 (Reference Konakovic Lukovic et al. 2020): Many thanks for pointing this out, we have now added this reference. All the algorithms used in our comparison were implemented using the same set of python libraries for both the modelling and acquisition. The DGEMO implementation uses different dependencies, which we believe will result in an unfair comparison. Nevertheless, a recent benchmark by Daulton et al (2021) showed that DGEMO has comparable or weaker performance on the standard hypervolume compared to qNEHVI and qNParEGO, which we did compare with.
> >
> > - Q6 (error bars): We have now restructured the plots and added error bars everywhere.
> >
> > - Q7 (performance): In the many of the test cases our method was among the best performers in terms of the standard hypervolume. The few cases where this did not occur (e.g. Penicillin example) was when specific parts of the Pareto frontier were isolated. The improvements the method offers are indeed marginal, but this is over many methods and on a variety of problems that had received significant attention in the past.
> >
> > - Q8 (Penicillin): The Penicillin objective function is obtained by solving a set of differential equations using a discretization scheme. As a result, the change in the hypervolume of the proposed points varies in a noisy manner.
> >
> > - Q9 (single-objective plots): Figures 1, 6-10 are there just to aid in terms of intuition, so we opted for the simple choice of single objective. We agree this intuition should be used with caution in the multi-objective case, but we still believe that the plots can be useful to illustrate some differences between MES, PES, and JES.
> >
> > - Q10 (several metrics): We completely agree and this was our motivation for including additional visualizations of the Pareto front and performance results over the different regions of the objective space (Appendix J5 and J6).
> >
> > Typos:
> >
> > All typos have been corrected, many thanks for spotting.

---

> > > ### Comment · Reviewer_ucDe · 2022-08-07
> > > **Response**
> > >
> > > Thank you for the detailed response.
> > >
> > > Most of my concerns were addressed, especially from a presentation perspective.
> > >
> > > The parts that I still disagree with/were not addressed are the following:
> > >
> > > * The acronyms of different algorithm names would not be problematic in the general case; however, in this specific case, PES and MES actually refer to two famous acquisition functions for single objective optimization, so I still think this should be changed across the paper. Adding two letters for each acronym will not complicate anything and will serve the context. This is especially true for this paper, where there are plots involving single objective experiments.
> > >
> > > * DGEMO was compared to qNEHVI only in the noisy setting while it was not built to support the noisy setting. This means that it is not expected to perform well. This does not mean that it will not outperform other algorithms in the non-noisy setting. DGEMO is a very strong baseline, recent and relevant. I believe it is very common to compare methods that are not necessarily built with the same library, most papers do not reimplement other baselines from scratch.
> > >
> > >  >  The algorithm is valid for both the single-objective and multi-objective setting as can be a random vector-valued function with M components. This is registered at the beginning of Section 2.
> > >
> > > * I totally disagree with this statement. The inequality relationship in the case of a single objective and multi-objective are totally different. This has been already shown and discussed in the extension of MES to MESMO and PFES and also PES to PESMO. So assuming that this is straightforward is concerning.

---

> > > > ### Author Response · Authors · 2022-08-08
> > > > **Response**
> > > >
> > > > Thank you for your reply. We will further clarify the difference between the single-objective and multi-objective problem in the paper. As pointed out by the reviewer, we use the Pareto partial ordering in the multi-objective setting and the standard total ordering in the single-objective setting. Computationally, the main difference occurs when computing the region of integration. Specifically, in the single-objective setting, the box-decomposition of the dominated region (Step 4 of Alg 1) is (-infty, y*], where y* is the maximum.

---

### Official Review · Reviewer_XDbZ · 2022-07-24

**Rating:** 7
**Confidence:** 3
**Soundness:** 3 good
**Presentation:** 3 good
**Contribution:** 3 good

**Summary:**

In this paper, a Bayesian optimization algorithm using Joint Entropy Search (JES), a new entropy search type acquisition function, is proposed to efficiently estimate the Pareto solution set for multi-objective optimization problems. JES is defined as the mutual information between the pair of optimal solution set and the optimal value set,  (X^*, Y^*) and the new query point (x, y). While PES and MES, the main conventional entropy search type acquisition functions, focus only on the marginal distribution of the optimal solution and optimal value, respectively, JES determines the next acquisition point by considering their joint distribution.

JES can be expressed as the difference of two differential entropies as in other entropy search type acquisition functions. For its optimization, the authors proposed a method to approximate the expected value of entropy for the pair (X^*, Y^*) of the optimal solution set and the optimal value set that appears in the second term of the JES acquisition function. Specifically, when there is no observation noise, entropy can be calculated analytically since the argument distribution is a multivariate truncated normal distribution, and when there are observation error, the entropy of the multivariate normal distribution that bounds it from above can be used as an approximation. Specifically, when there is no observation noise, entropy can be calculated analytically since the argument distribution is a multivariate truncated normal distribution. When there are observation errors, a multivariate normal distribution whose 1st and 2nd moments are equivalent to the argument distribution can be used as an approximation. This is because the entropy of latter distribution is bounded above by the entropy of former one.

The proposed method, the JES acquisition function, is also extended to the batch optimization setting. Specifically, the authors proposed a method in which the entropy of the joint distribution for the batch points is bounded from above by the sum of entropies for each point, and the latter is used to select each point independently.

In the experiments, the proposed method is compared with existing multiobjective Bayesian optimization methods for different optimization problems with 2-, 3-, and 4-objectives, using the difference of hypervolumes as an evaluation criteria.

**Questions:**

- As pointed out in Weakness, the assumption that Y^* is a finite set, as placed in Lemma 1, seems to be a strong one; it seems to be the key lemma for the approximate computation of the JES acquisition function, but is it not an issue?

- As pointed out in weakness, since the proposed method decomposes all computations into point-by-point computations, I think that it needs to be carefully evaluated in terms of computational cost compared to existing methods. Have you done any comparison of actual computational costs?

**Limitations:**

The authors adequately addressed the limitations and potential negative societal impact of their work.

**Strengths And Weaknesses:**

Strengths

- The proposed new entropy-search type acquisition function, JES, considers the joint distribution of optimal position x^* and optimal value y^*, and is expected to provide more information about the optimal solution than PES, which considers only the former, or MES, which considers only the latter.

- Different computation methods have been proposed for the entropy of the second term in JES, with and without noise. Both are derived in an analytic form, and the advantage is that costly numerical computations can be avoided.

- As a method for obtaining batch points, the joint conditional distribution for a batch is upper bounded by the sum of the conditional distributions for each point, avoiding the intractable computations of the former. With respect to the goodness of this upper bound, the authors show that by showing the submodularity of the acquisition function, an iterative method of sequential acquisition at each point achieves a regret upper bound of e^-1.

- While the hypervolume of the Pareto set (e.g., its expected reduction), which is often used as an acquisition function in multi-objective optimization, is not invariant with respect to monotonic transformations of the output, the proposed method, JES, is shown to be invariant to the same transformation.

Weakness

- The assumption is made in Lemma 1 that the true Pareto optimal value set Y^* is a finite set, which appears to be a strong condition given that in general there are an infinite number of Pareto solutions.

- The computations of the conditional distribution for the Pareto set and the computations for acquiring batch points are each broken down into a computations for each point, which is expected to become a computationally expensive method when the number of elements in the Pareto solution set increases as iteration progresses.

---

> ### Author Response · Authors · 2022-08-01
> **To XDbZ**
>
> - It unavoidable that numerically the Pareto frontier approximation will be be constructed on a finite set of points. This is a standard assumption in the literature and we believe is also sensible given the interest here is on approximations with finite number of observations. In addition, in Appendix J.3 our sensitivity analysis suggests that there is no visible improvement as the number of points approximating the Pareto frontier increase beyond a moderate number of points (around 10 points for ZDT2 with 2 objectives on 6 dimensions).
>
> - Regarding computational cost increase w.r.t number of points, this increases moderately when computations are performed in sequence on one processor (Figure 15). Notably, some of these calculations can benefit from parallelization in order to get a speed-up.

---

> > ### Comment · Reviewer_XDbZ · 2022-08-09
> > **Thank you for your response**
> >
> > Thank you for your detailed answer to my questions.
> > My concern was whether the assumption that Y is a finite set affect the practical performances, but I was convinced that experiments have already confirmed that this is not the case in practice.
> > After reconsidering the comments I have received, together with the comments of the other reviewers, I would like to increase the score by one.

---

### Author Response · Authors · 2022-08-01
**All reviewers**

We thank all the reviewers for their comments and the effort to assess our paper. In our revised manuscript, we have addressed the most common and important points:

- We have clarified that the modification to the noiseless conditional entropy estimate was meant as an ad hoc adjustment to account for the amount of exploration when there is observation noise. As this result caused some confusion, we have relegated it to Appendix D in favour of emphasizing our main result in Prop. 2 (previously Prop. 3), which gives a more principled and better performing estimate.

- We have further emphasised the connection to existing work in the main text.

- We have provided error bars in the plots.

- We have added an in-sample standard hypervolume comparison (Appendix J.4) as requested by one of the reviewers.

- We have augmented the discussion of the experiments with additional points raised by the reviewers.

- We have sorted all typos and minor errors.

Below we provide a point-by-point response to each reviewer separately.

---

### Meta-Review · Area_Chair_Luur · 2022-08-26

**Recommendation:** Accept
**Confidence:** Certain

**Metareview:**

The authors propose an entropy search method for multi-objective Bayesian optimization that considers the mutual information gain of the location and value of the optimizer simultaneously while selecting query points.  Most reviewers found the approach to be interesting.  The work is commendable in its attempt to rigorously compare different information-based acquisition functions via high-quality re-implementations of algorithms (e.g., PESMO).  Many reviewers left detailed comments that the authors addressed in part, or agreed to examine in the camera ready.  Examples include more rigorous examination of the performance with respect to noise and batch size, inferential statistics for the experiments, and transparent reporting on the strengths and weaknesses relative to the existing literature.

**Award:**

No

---

### Decision · Program_Chairs · 2022-09-14

Accept